# Uncertainty Analysis of Two Copula-Based Conditional Regional Design Flood Composition Methods: A Case Study of Huai River, China

**Shiyu Mou [1]**, **Peng Shi [1,2]**, **Simin Qu [1,*]**, **Xiaomin Ji [3]**, **Lanlan Zhao [4]**, **Ying Feng [1]**, **Chen Chen [1] and Fengcheng Dong [1]**

[1] College of Hydrology and Water Resources, Hohai University, Nanjing 210098, China; ariesmsy@hhu.edu.cn (S.M); ship@hhu.edu.cn (P.S); 171301010009@hhu.edu.cn (Y.F); Johnnychenhhu@gmail.com (C.C); dfcheng93@126.com (F.D)

[2] State Key Laboratory of Hydrology-Water Resources and Hydraulic Engineering, Hohai University, Nanjing 210098, China

[3] Department of Water Resources Evaluation, Jiangsu Province Hydrology and Water Resources Investigation Bureau, Nanjing 210029, China; 13512530598@163.com

[4] Bureau of Hydrology, Ministry of Water Resources of the People's Republic of China, Beijing 100053, China; zhaolanlan@mwr.gov.cn

\* Correspondence: wanily@hhu.edu.cn; Tel.: +86-138-5171-3291

**Abstract:** The issue of regional design flood composition should be considered when it comes to the analysis of multiple sections. However, the uncertainty accompanied in the process of regional design flood composition point identification is often overlooked in the literature. The purpose of this paper, therefore, is to uncover the sensibility of marginal distribution selection and the impact of sampling uncertainty caused by the limited records on two copula-based conditional regional design flood composition methods, i.e., the conditional expectation regional design flood composition (CEC) method and the conditional most likely regional design flood composition (CMLC) method, which are developed to derive the combinations of maximum 30-day flood volumes at the two sub-basins above Bengbu hydrological station for given univariate return periods. An experiment combing different marginal distributions was conducted to explore the former uncertainty source, while a conditional copula-based parametric bootstrapping (CC-PB) procedure together with five metrics (i.e., horizontal standard deviation, vertical standard deviation, area of 25%, 50%, 75% BCIs (bivariate confidence intervals)) were designed and employed subsequently to evaluate the latter uncertainty source. The results indicated that the CEC and CMLC point identification was closely bound up with the different combinations of univariate distributions in spite of the comparatively tiny difference of the fitting performances of seven candidate univariate distributions, and was greatly affected by the sampling uncertainty due to the limited observations, which should arouse critical attention. Both of the analyzed sources of uncertainty increased with the growing T (univariate return period). As for the comparison of the two proposed methods, it seemed that the uncertainty due to the marginal selection had a slight larger impact on the CEC scheme than the CMLC scheme; but in terms of sampling uncertainty, the CMLC method performed slightly stable for large floods, while when considering moderate and small floods, the CEC method performed better.

**Keywords:** regional design flood composition; GH copula function; uncertainty analysis; Huai River basin

## 1. Introduction

Design flood analysis provides reasonable hydrological design values for water conservancy and wading projects [1,2]. When it comes to the analysis and calculation of multiple sections, it is necessary to deal with the issue of the regional design flood composition. Traditionally, the basic access of solving this issue includes two steps. First, the search for proper combinations of natural flood variables that occurred at different sub-basins above the study section. Second, the derivation of the design flood hydrograph at each sub-basin by amplifying the typical flood hydrograph using the design values calculated in the first step [3]. Hence, the selection of proper regional design flood composition schemes is of vital importance. In general, regional design flood composition is a spatially stochastic issue, and the most scientific and rational method to describe this law of nature is to build the joint probability distribution of flood variables in each sub-basin.

In recent years, the copula function has been extensively used in establishing joint distributions of relevant hydrological variables since being introduced in the hydrology and geosciences domain in 2003 [4]. For instance, frequency analysis describing extreme events, like rainfall, rainstorms, floods, or droughts [5–9], return period analysis [10,11], multivariate simulation [12,13], risk assessment [14,15], and some new insights for multivariate design quantile estimation [16–18]. Previous studies have indicated that the copula function was an effective tool in multivariate analysis, owing to the characters of integrating arbitrary complex marginal distributions and simulating the nonlinear relationship between any number variables. Thus, several studies using copula methods to derive regional design flood composition have been conducted. Yan et al. [19] first introduced the copula function to the composition of a single-reservoir flood control system and proposed two representative regional composition schemes: Conditional expectation regional design flood composition (CEC) and the most likely regional design flood composition (MLC). Liu et al. [20] derived the general formula of the MLC method, which had sufficient statistical basis and strong operability in practical calculations. Li et al. [21] improved the traditional discrete summation method by discretizing the conditional probability curve with the copula function. By contrasting with the traditional equivalent frequency regional composition (EFC) [22] scheme, Guo et al. [23] proved the satisfactory performance of the CEC and MLC methods. Learning from the algorithms of Yan et al. and Guo et al., the present study takes full advantage of the good performances of the MLC and CEC methods, and subsequently develops the conditional most likely regional composition (CMLC) method.

It is worth noting that any estimation comes along with uncertainty, and extensive uncertainties exist in multivariate frequency analysis. With the copula-based methodology for multivariate design realization estimation, a few scholars have studied the taxonomies of accompanying uncertainty, and have discussed different sources and kinds of uncertainty. Serinaldi [24] proposed three algorithms for performing multivariate risk analysis under sampling uncertainty. Dung et al. [25] developed a non-parametric bootstrapping procedure for investigating the uncertainty of the parameter estimation method, model selection, and sampling, and the results further revealed that compared with sampling uncertainty, the other two sources of uncertainty were of less importance. However, the impact of uncertainty due to the selection of marginal distributions has not been sufficiently analyzed in previous studies.

The extensive uncertainties not only have great influence on hydrometeorology event identification or Q-V (flood peak and corresponding flood volumes) combinations, but also have impacts on the conditional regional design flood composition estimation. The first objective of this study was to introduce two copula-based conditional regional design flood composition methods (i.e., CEC and CMLC), which considered the spatial correlation of regional floods, and the second objective aimed at revealing the sensibility of marginal distribution selection and the effect of sampling uncertainty caused by the limited records on two proposed composition methods. The two main sources of uncertainty, particularly sampling uncertainty, are frequently neglected in the literature despite their well acknowledged importance, possibly because of their technical difficulty [26]. Bengbu hydrological station in the middle of the Huai River basin was selected as a case study.

This research provides new approaches for analysis of regional design flood composition and useful information to clarify flood risk.

## 2. Methodology

The objective of this paper is twofold; first, to introduce two copula-based conditional Regional design flood composition methods and, second, to reveal the impact of two uncertainty sources on the proposed method. Figure 1 presents the methodological framework of this paper, and in the following part of this section, we elaborate on the approach proposed and employed in this framework, including the theory of copula, theory of the regional design flood composition method, and the procedure and metrics for investigating the uncertainty.

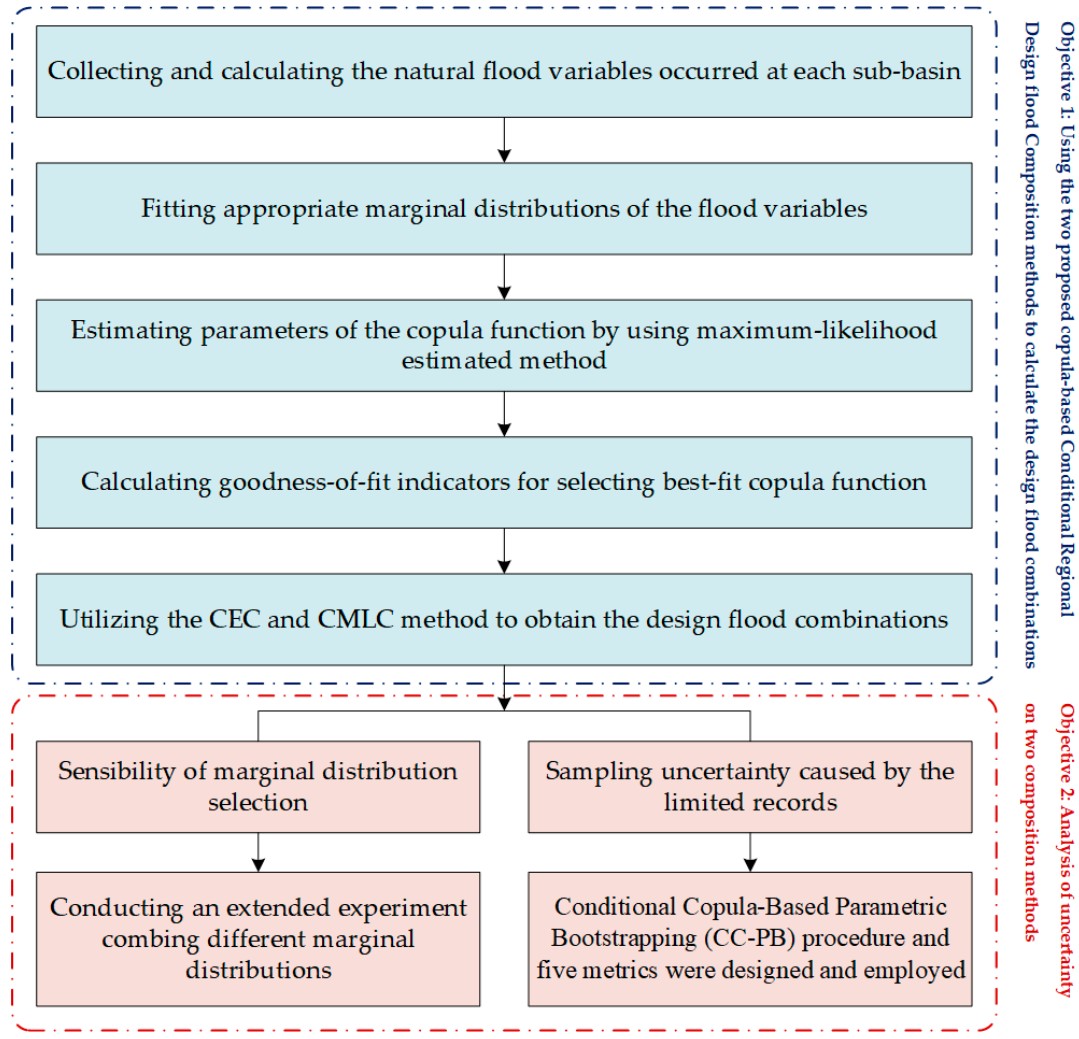

**Figure 1.** Flowchart of the proposed methods in this paper.

### 2.1. Copula Theory

A copula is a function connecting the multivariate probability distribution to its marginal ones. Better understood, it is a family of multivariate probability distributions whose margins are standard uniform distributions. For a bivariate case, the random variables, $X$ and $Y$, respectively denote the horizontal and vertical directions in $R^2$. $F_X(x)$ and $F_Y(y)$ are the cumulative distribution functions (C.D.Fs) of $X$ and $Y$, respectively, which are uniformly distributed random variables on [0,1] and are respectively denoted as $u$ and $v$. According to Sklar's Theorem [27], a two-dimensional copula can be represented as:

$$F_{X,Y}(x,y) = C[F_X(x), F_Y(y); \theta] = C(u,v;\theta) \tag{1}$$

where $F_{X,Y}(x,y)$ denotes the bivariate C.D.F of $X$ and $Y$; and $C(u,v;\theta)$ represents the unique bivariate copula function with the parameter, $\theta$.

Among the copula families, the Archimedean family is widely used for hydrological analysis because of its simplicity and general properties [28–31]. Copulas that belong to this family provide analytically tractable models and possess different desirable characteristics. It is found that annual maximum (AM) flood volumes are extreme events of interest and are usually variables with upper tail dependence. Another set of extreme events that would be of interest (though not considered in this study) comprise peak volumes occurring in an independent and identically distributed way or the peak-over-threshold events, the extraction of which follow the procedure like that found in Onyutha [32]. The Gumbel-Hougaard (GH) copula is the only Archimedean copula in the list of the multivariate extreme value (MEV) copula family, which is beneficial to deal with flood risk [25,29]. Therefore, three types of GH copula [28,30,31], i.e., symmetric GH copula, two-para GH copula, and the asymmetric GH copula, were applied in this study, and the latter of which is constructed into a three-parameter version with Marshall-Olkin copulas on the boundaries. The corresponding copula functions and parameters for the three GH copulas are further described in Table 1.

**Table 1.** Summary of the three candidate GH copula functions.

| Name | Descriptions |
|------|--------------|
| Symmetric GH copula | $C(u,v;\theta) = \exp\left\{ -\left[ (-\log u)^\theta + (-\log v)^\theta \right]^{1/\theta} \right\}$, $\theta \in [1, +\infty)$ |
| Two-para GH copula | $C(u,v;\beta_1,\beta_2) = \left\{ \left[ \left( u^{-\beta_2} - 1 \right)^{\beta_1} + \left( v^{-\beta_2} - 1 \right)^{\beta_1} \right]^{1/\beta_2} + 1 \right\}^{-1/\beta_2}$, $\beta_1 \in [1, +\infty)$, $\beta_2 \in [0, +\infty)$ |
| Asymmetric GH copula | $C(u,v;\theta,\pi_2,\pi_3) = \exp[-A(-\log u, -\log v; \theta, \pi_2, \pi_3)]$, $A(x,y;\theta,\pi_2,\pi_3) = \left[ (\pi_2 x)^\theta + (\pi_3 y)^\theta \right]^{1/\theta} + (1-\pi_2)x + (1-\pi_3)y$, $\theta \in [1, +\infty)$, $\pi_2 \in (0,1)$, $\pi_3 \in (0,1)$ |
| | $u \in [0,1], v \in [0,1]$ <br> $\theta, \beta_1, \beta_2, \pi_2, \pi_3$: Copula parameter <br> $C$: Copula function <br> $A$: *Marshall-Olkin* copulas |

*2.2. Regional Design Flood Composition*

2.2.1. Basic Concept of Regional Design Flood Composition

Figure 2a shows the sketch map of a typical regional flood composition issue, in which A denotes the upstream section with water conservancy projects here; B denotes the interval sub-basin; and C denotes the downstream section. The flood variable at section C is the grand total of flood variables at section A and interval B according to the principle of water balance.

The aim of regional design flood composition, in a nutshell, is to seek for a proper combination of flood variables that occurred at the upstream section, A, and interval, B, by decomposing flood events that occurred at the downstream section, C. For the purpose of analyzing the flood-control effectiveness of different regional composition schemes, it is usually necessary to draw up a number of calculation schemes based on floods from different sub-basins. Therefore, the first step of regional design flood composition analysis mentioned above is of great significance, which acts as the focus of this study.

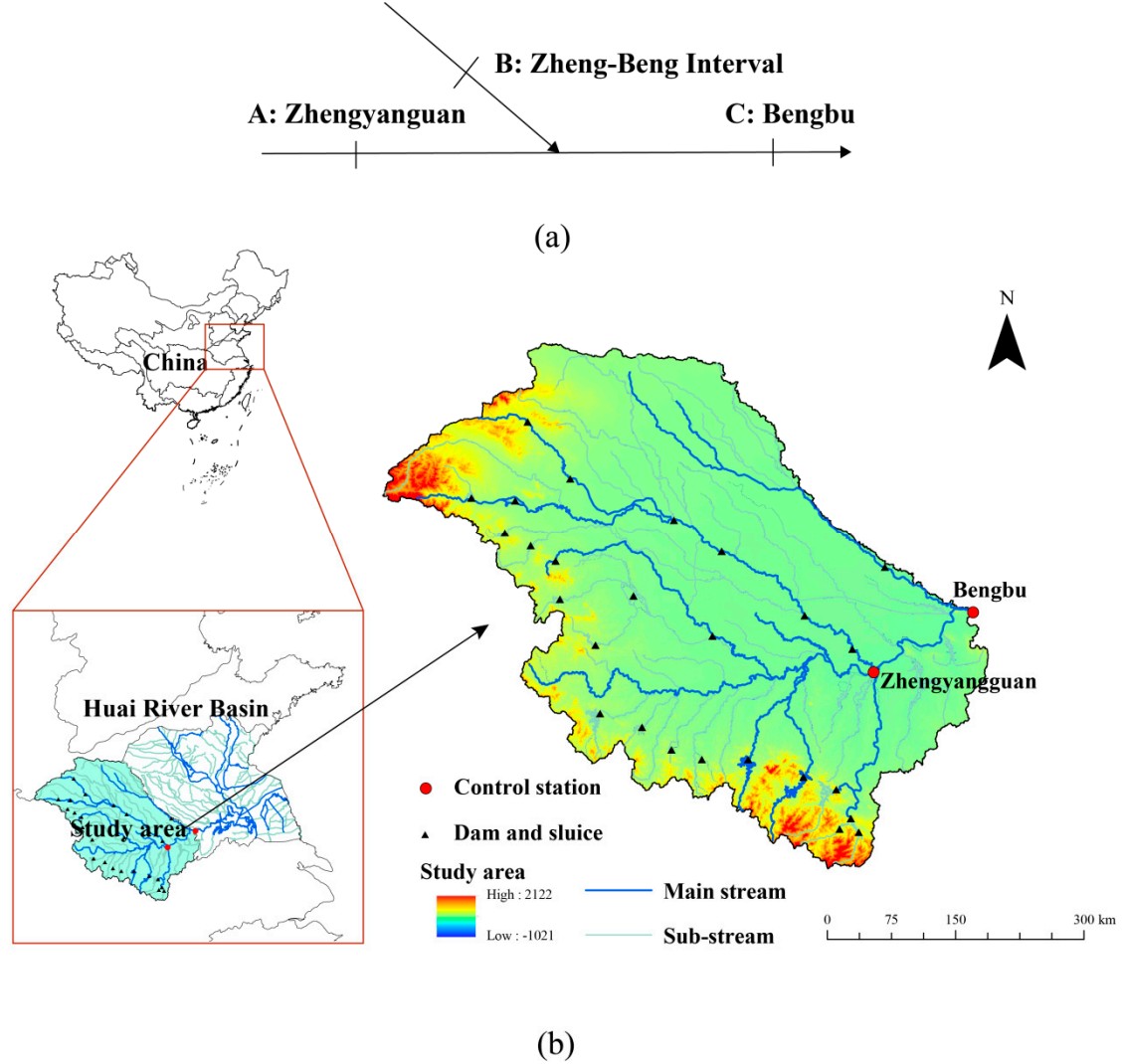

(a)

(b)

**Figure 2.** (**a**) Sketch map of a typical regional flood composition issue. (**b**) Location of Bengbu and Zhengyangguan hydrological stations in the Huai River basin.

### 2.2.2. Flood Regional Composition Methods Based on Copulas

Actually, not all the combinations are consistent with the inherent disciplines of hydrologic events development and the engineering design requirements. Based on the copula function, two representative regional design flood compositions are presented: i.e., CEC and CMLC, which have statistical basis and are able to reflect the inherent disciplines of hydrologic events.

When the value of the given flood variable, $X$, at the upstream section, A, is $x_p$, and the corresponding value ($y$) of the flood variable, $Y$, at the interval, B, is not fixed. A different y corresponds to a different probability of occurrence. The combination of $x_p$ and the conditional expected value of $Y$ ($E(y|x_p)$) is denoted as the conditional expectation regional design flood composition (CEC), which denotes the average level of different composition schemes. $E(y|x_p)$ can be expressed by Equation (2):

$$E(y|x_p) = \int_{-\infty}^{+\infty} yf(Y \leq y|X = x_p)dy = \int_{-\infty}^{+\infty} yc(u,v)f_Y(y)dy = \int_0^1 F_Y^{-1}(v)c(u,v)dv \qquad (2)$$

where $f(Y \leq y|X = x_p)$ is the conditional probability density function (P.D.F), and $f(Y \leq y|X = x_p) = c(u,v)f_Y(y)$ according to the copula theory; $c(u,v)$ is the density function of $C(u,v)$, and $c(u,v) = \partial^2 C(u,v)/\partial u \partial v$; $f_Y(y)$ is the P.D.F of $Y$; and $F_Y^{-1}(\cdot)$ is the inverse C.D.F of $Y$.

Equation (2) cannot find the analytical solution, which needs to be solved by numerical integration. Further, the CEC points $(x_p, E(y \mid x_p))$ will be obtained.

Based on the principle of maximizing the conditional density function, the formula of another copula-based composition method, namely the CMLC method, is derived. Given $y$ equals to the maximum value of conditional PDF $f(Y \leq y \mid X = x_p)$ $(y_M)$, the combination of $x_p$ and $y_M$ (i.e., $(x_p, y_M)$) is regarded as the conditional most likely regional design flood composition (CMLC).

The first order derivative of $f(Y \leq y \mid X = x_p)$ can be derived as:

$$\frac{df(Y \leq y \mid X = x^p)}{dy} = c_2 (f_Y(y))^2 + c(f_Y(y)) \tag{3}$$

where $c_2 = \partial c / \partial v$. According to the method of evaluation extremum in calculus, the point where the first order derivative equals to zero can make the original equation reach the maximum (minimum) value. Given Equation (3) equals to zero, after simplification, the following equation is supposed to be satisfied:

$$c_2 \cdot f_Y(y) + c \cdot (f_Y{}'(y)/f_Y(y)) = 0 \tag{4}$$

where $f_Y{}'(y)$ are the corresponding derived functions of PDF $f_Y(y)$.

$(x_p, y_M)$ can be obtained by solving the nonlinear Equation (4) with the Newton iteration method [33]. Similarly, when the interval flood volumes are taken as the condition, we also can get two corresponding regional design flood composition schemes.

By using the CEC and CMLC method to obtain the combination $(x, y)$, the value of $Z$ at the downstream site, C, can be easily calculated.

Let the conditional C.D.F $F(Y \leq y \mid X = x_p)$ be equal to $\alpha/2$ and $1 - \alpha/2$, respectively, where $\alpha$ is the significance level, then the corresponding value of the flood variable, $Y$, at the interval sub-basin, B, can be described as $y_L$ and $y_U$, respectively. $[y_L, y_U]$ is regarded as the corresponding confidence interval (CI) given $X = x_p$, which can quantitatively evaluate the uncertainty associated with the composition schemes estimation and provide a basis to clarify the flood risk. The CEC and CMLC methods together are known as the conditional regional composition method.

### 2.3. Conditional Copula-Based Parametric Bootstrapping (CC-PB) Procedure

To explain the sampling uncertainty, a conditional copula-based parametric bootstrapping (CC-PB) procedure was designed as follows:

1.  Fit the marginal distributions and parametric copula function for the original dataset (i.e., $X$ and $Y$). The parameters of the chosen marginal distributions and copula function are estimated by the L-moment method and maximum pseudo-log-likelihood (ML) method, respectively.
2.  Predefine $N_B$ bivariate bootstrapping samplings of size $n$ through the usage of the conditional simulation method [29], and then obtain $Z^* = (X^*, Y^*) = (x_{ij}, y_{ij})$ from the bivariate dependence structure via the probability integral transform (PIT) using the fitted parameters of the margins $(i = 1, \ldots, n; j = 1, \ldots, N_B)$.
3.  Estimate the parameters of marginal distributions and the parametric copula function of $Z^*$ utilizing the same estimation method used for the original dataset, then obtain $N_B$ pairs of $F_j{}^*(x_{ij}, y_{ij})$, $(i = 1, \ldots, n; j = 1, \ldots, N_B)$.
4.  Identify the CEC and CMLC realizations for different $(x, y)$ pairs by Equations (2) and (4), respectively.
5.  Utilize these realizations to estimate the bivariate confidence intervals (BCIs) by employing the kernel density estimation (KDE) method [34].

*2.4. Metrics for Sampling Uncertainty*

Yin et al. [35] proposed four evaluation indexes to quantify the uncertainty of bivariate quantile estimation. Similar to the idea of Yin et al., five metrics, i.e., horizontal standard deviation, $\sigma_x$; vertical standard deviation, $\sigma_y$; area of 25% BCI, $S_{25\%}$; area of 50% BCI, $S_{50\%}$; and area of 75% BCI, $S_{75\%}$, are utilized to quantify the sampling uncertainty. $S_{25\%}$, $S_{50\%}$, and $S_{75\%}$ are highly dependent on the grids contained in the BCIs and are approximated by Riemann sums using the R-package 'ks' [34]. The other two metrics are calculated by the following equations:

$$\sigma_x = \sqrt{\frac{1}{N_B} \sum_{i=1}^{N_B} \left(W_{x_i} - \hat{W}_x\right)^2} \tag{5}$$

$$\sigma_y = \sqrt{\frac{1}{N_B} \sum_{i=1}^{N_B} \left(W_{y_i} - \hat{W}_y\right)^2} \tag{6}$$

where $W_{x_i}$ and $W_{y_i}$ represents the estimated value of the flood variable at each sub-basin for the $i$ simulated dataset, respectively; and $\hat{W}_x$ and $\hat{W}_y$ is the expectation design value of the flood variable at each sub-basin, respectively.

## 3. Study Area and Data

The proposed method was used in Bengbu hydrological station, located in the middle of the Huai River Basin (30°55′–36°36′ N, 111°55′–121°25′ E). The study area refers to the upper–middle reach region of the Huai River Basin above the Bengbu section, with an area of $12.13 \times 10^4$ km$^2$, including the mainstream and five major tributaries, still termed as the Huai River Basin (Figure 2b). According to topographic relief and river characteristics, the middle of Huai River can be divided into two interval sections above and below the Zhengyangguan section. The drainage area above the Zhengyangguan section is $8.86 \times 10^4$ km$^2$, and the length of the stream from Zhengyangguan to the Bengbu section is 119 km, which drains an area of $3.27 \times 10^4$ km$^2$.

At present, most rivers in the upper and middle reaches of Huai River Basin are regulated by a mass of dikes, floodgates, and large and medium-sized reservoirs [36]. It is of great significance to analyze the regional design flood composition at the Bengbu section, regarding the Zhengyangguan section as the boundary. A sketch diagram of this area is also shown in Figure 2a, which can be generalized as a typical regional design flood system. The daily flood series have been systematically recorded at Zhengyangguan and Bengbu hydrological stations for 63 years (1953–2015). The natural flow process of large reservoirs and lakes, the amount of intercepting water in flood diversion regions, and irrigation reference water were calculated for the Zhengyangguan and Bengbu section, and the measured flow processes were added to these sections to obtain the natural flow processes of corresponding sections. For a large or medium-scale water conservancy project, as a rule, the flood routing is controlled by the flood volume, so the regional design flood composition generally refers to the combination of flood volumes for convenience [23,37]. According to the flood characteristics of the Huai River basin and the regulation characteristics of water conservancy projects, the designed flood control period is selected for 30 days [22]. Therefore, annual maximum 30-day flood volume data ($W_{30d}$) from Zhengyangguan Station and Bengbu Station were collected. The natural flood process of the Zhengyangguan section was routed to the Bengbu section with the Muskingum method [38]. By deducting this routed flood process from the natural flood process of the Bengbu section, we could obtain the flood process of the interval basin between Zhengyangguan and Bengbu (hereinafter referred to as the Zheng-Beng interval). Then, the Zheng-Beng interval, $W_{30d}$, could be easily obtained.

Figure 3a shows the Zhengyangguan $W_{30d}$ and Zheng-Beng interval $W_{30d}$ series. These two series exhibited similar changing variations throughout the 63-year data. The dispersion of these two series was measured by boxplot graphs (Figure 3b), and the symmetry of the boxplot implies that there is a

certain level of right skew for both series. Figure 3c illustrates that Zhengyangguan $W_{30d}$ has a rough linear correlation with the Zheng-Beng interval $W_{30d}$, and the Pearson correlation coefficient between the $W_{30d}$ of the two sub-basins is 0.869, which connotes a comparatively marked linear relationship. Nevertheless, the dependence structure of the dataset is supposed to be more complicated than a straightforward linear correlation, which can be more flexibly analyzed based on copulas. Table 2 lists some basic statistical parameters of the two series, which describe the structure and overall situation of the data.

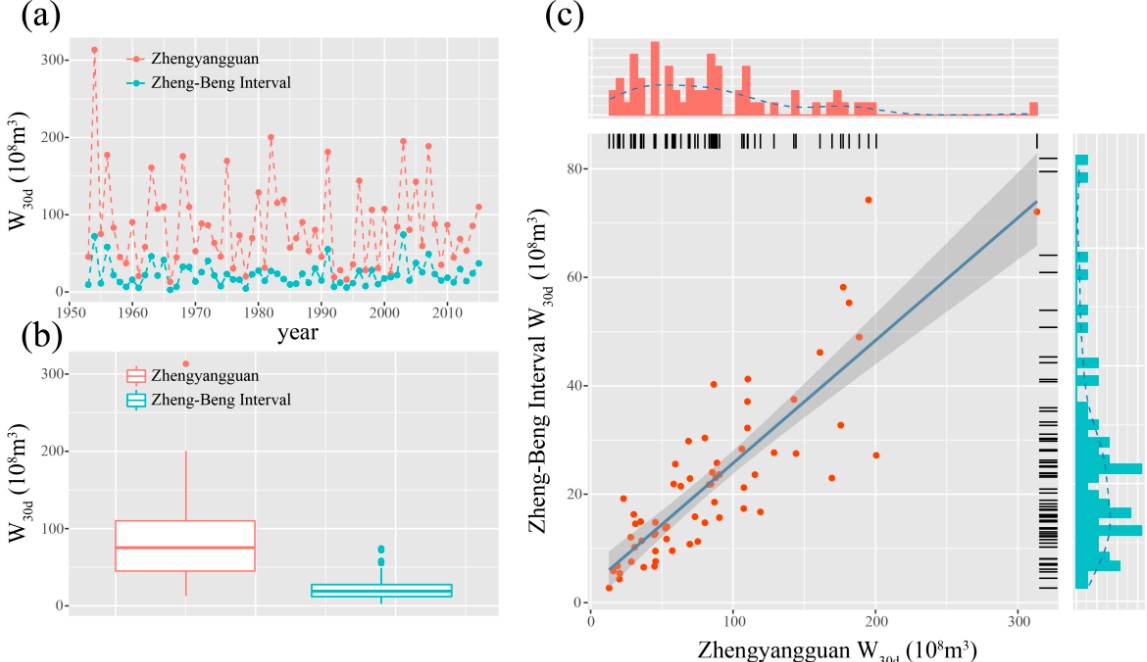

**Figure 3.** (**a**) Zhengyangguan $W_{30d}$ and Zheng-Beng interval $W_{30d}$ series. (**b**) Boxplot graph of Zhengyangguan $W_{30d}$ and Zheng-Beng interval $W_{30d}$ series. (**c**) Scatter plot of Zhengyangguan $W_{30d}$ and Zheng-Beng interval $W_{30d}$ together with linear regression line and histogram of marginal distributions on the side panels.

**Table 2.** Statistics of Zhengyangguan and Zheng-Beng interval $W_{30d}$.

|  | Zhengyangguan $W_{30d}$ ($10^8$ m$^3$) | Zheng-Beng Interval $W_{30d}$ ($10^8$ m$^3$) |
|---|---|---|
| [Min, Max] | [12.95, 313.47] | [2.72, 74.29] |
| Median | 75.27 | 19.22 |
| Mean | 85.76 | 22.51 |
| Standard deviation | 57.75 | 15.39 |
| Skewness | 1.32 | 1.45 |
| Kurtosis | 2.22 | 2.11 |
| Interquartile range | 65.04 | 15.66 |

## 4. Results and Discussions

### 4.1. Selection of Marginal Distribution

Sklar's theorem is intended to separate the multivariate model into two independent parts, i.e., the fitting of the marginal C.D.Fs and the selection of a suitable parametric copula. Therefore, in order to construct the bivariate model for $W_{30d}$ at each sub-basin, the first step is to choose appropriate marginal distributions. Seven widely-used distributions in hydrology [37,39,40], namely Pearson type III (PE3), three parameters log-normal (LN3), generalized extreme value (GEV), generalized pareto (GP), gamma (GAM), gumbel (GUM), and generalized logistic (GLO), were picked

as candidate distributions for flood volume. Figure 4 illustrates the distributions of the $W_{30d}$ series in each study sub-basin fitted by the seven candidate distributions. The quantile-quantile plot, C.D.F plot, and P.D.F plot are exhibited in this figure.

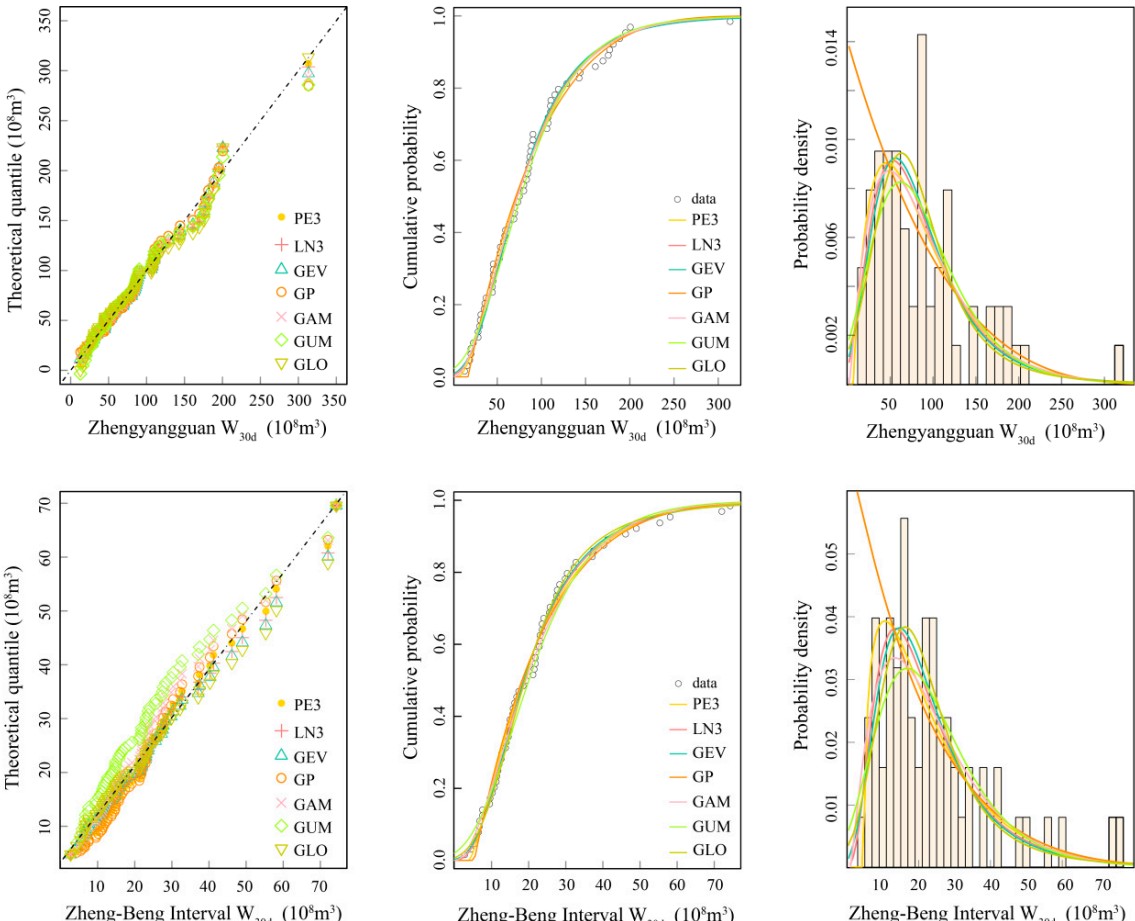

**Figure 4.** Fitting distributions to Zhengyangguan $W_{30d}$ and Zheng-Beng interval $W_{30d}$: The quantile-quantile plot (**left**), C.D.F plot (**middle**), and P.D.F plot (**right**).

Figure 4 demonstrates that seven theoretical distributions basically fit the observed data. To select the best-fitting distribution, we compared the performance among the distributions through their root-mean-square error (RMSE) and Akaike information criteria (AIC) [41] values. The goodness of fit (GOF) with the aid of the Cramér–von Mises (CvM) test also performs to provide support to select the candidate distributions for the $W_{30d}$ series at each sub-basin. The Cramér–von Mises statistic ($w^2$) is designed to quantify the distance between the empirical and fitted distributions [42]. Lower $w^2$ values and higher *p*-values mean a preferable imitative effect.

Table 3 shows the parameter estimation results of the candidate distributions using the L-moment method, the RMSE, AIC values, and Cramér–von Mises test results are also presented. It can be seen that the PE3 is identified as the best fitting distribution function for Zhengyangguan $W_{30d}$ data, yielding the minimum RMSE and AIC values, while MEV is the best fitting one for $W_{30d}$ at the Zheng-Beng interval. All of the *p* values listed in Table 3 are quite bigger than 0.05, indicating that seven theoretical distributions are capable of fitting the distributions of $W_{30d}$ at each sub-basin at the 5% significance level.

**Table 3.** Parameters estimated by the L-moment method and GOF of marginal distributions.

| Region | Series | Functions | Parameters | | CvM Test | | RMSE | AIC |
|---|---|---|---|---|---|---|---|---|
| | | | Name | Estimated Value | $w^2$ | $p$ | | |
| Zhengyangguan | $W_{30d}$ | PE3 | $[\alpha, \beta, \gamma]$ | [1.921, 0.024, 4.870] | 0.027 | 0.985 | 2.881 | 668.32 |
| | | LN3 | $[\mu_{\log}, \sigma_{\log}, \zeta]$ | [4.599, 0.497, −26.673] | 0.030 | 0.978 | 3.003 | 672.85 |
| | | MEV | $[\xi, \alpha, \kappa]$ | [58.042, 40.019, −0.105] | 0.034 | 0.964 | 3.013 | 674.06 |
| | | GP | $[\xi, \alpha, \kappa]$ | [16.994, 84.403, 0.227] | 0.042 | 0.924 | 3.214 | 673.34 |
| | | GAM | $[\beta, \alpha]$ | [39.100, 2.193] | 0.028 | 0.983 | 3.085 | 669.58 |
| | | GUM | $[\xi, \alpha]$ | [60.053, 44.544] | 0.051 | 0.869 | 3.324 | 674.90 |
| | | GLO | $[\xi, \alpha, \kappa]$ | [73.944, 28.046, −0.239] | 0.049 | 0.887 | 3.119 | 676.88 |
| Zheng-Beng Interval | $W_{30d}$ | PE3 | $[\alpha, \beta, \gamma]$ | [1.434, 0.077, 3.899] | 0.037 | 0.949 | 2.138 | 500.80 |
| | | LN3 | $[\mu_{\log}, \sigma_{\log}, \zeta]$ | [3.064, 0.579, −2.801] | 0.024 | 0.993 | 2.084 | 489.11 |
| | | MEV | $[\xi, \alpha, \kappa]$ | [15.037, 9.770, −0.161] | 0.023 | 0.994 | 2.025 | 488.53 |
| | | GP | $[\xi, \alpha, \kappa]$ | [5.366, 19.379, 0.130] | 0.060 | 0.815 | 2.355 | 501.80 |
| | | GAM | $[\beta, \alpha]$ | [10.109, 2.227] | 0.040 | 0.936 | 2.651 | 499.99 |
| | | GUM | $[\xi, \alpha]$ | [15.811, 11.614] | 0.069 | 0.762 | 3.170 | 504.70 |
| | | GLO | $[\xi, \alpha, \kappa]$ | [18.972, 7.066, −0.278] | 0.028 | 0.982 | 2.369 | 503.51 |

Note: P.D.F of PE3: $f(x) = \frac{\beta^\gamma}{\Gamma(\gamma)}(x-\alpha)^{\gamma-1}\exp(-\beta(x-\alpha))$, parameters: $\alpha$ (location), $\beta$ (scale), and $\gamma$ (shape, $\gamma > 0$); P.D.F of LN3: $f(x) = \frac{1}{(x-\zeta)\sigma_{\log}\sqrt{2\pi}}\exp\left(-\frac{(\ln(x-\zeta)-\mu_{\log})^2}{2\sigma_{\log}^2}\right)$, $\mu_{\log}$ (location), $\sigma_{\log}$ (scale), and $\zeta$ (lower bounds); C.D.F of GEV: $F(x) = \exp(-\exp(\kappa^{-1}\ln(1-\frac{\kappa(x-\xi)}{\alpha})))$, $\xi$ (location), $\alpha$ (scale, $\alpha > 0$), and $\kappa$ (shape, $\kappa > -1$); C.D.F of GP: $F(x) = 1-\exp(\kappa^{-1}\ln(1-\frac{\kappa(x-\xi)}{\alpha}))$, $\xi$ (location), $\alpha$ (scale), and $\kappa$ (shape); C.D.F of GAM: $F(x) = \frac{\beta^{-\alpha}}{\Gamma(\alpha)}\int_0^x t^{\alpha-1}\exp(-t/\beta)dt$, $\alpha$ (scale), and $\beta$ (shape); C.D.F of GUM: $F(x) = \exp(\exp(-\frac{x-\xi}{\alpha}))$, $\xi$ (location), $\alpha$ (scale); C.D.F of GLO: $F(x) = 1/(1+\exp(-\kappa^{-1}\log(1-\frac{\kappa(x-\xi)}{\alpha})))$, $\xi$ (location), $\alpha$ (scale), and $\kappa$ (shape).

## 4.2. Copula Function Construction

Three types of GH copulas, i.e., symmetric GH copula, two-para GH copula, and asymmetric GH copula, were selected as candidates to establish the bivariate model of the $W_{30d}$ series at Zhengyangguan and the Zheng-Beng interval. The contour density plots of three GH copulas (Figure 5) indicate that GH copulas have the advantage to simulate tail dependence, which may be weaker when displayed through the observed dataset. It is proven that improper selection of copula function types will bring significant uncertainty to the estimation of model simulation sequence design values [24,25]. Therefore, selecting and adjusting the best-fit copula function is a decisive step in the fitting process. Table 4 shows the result of the parameters of the three GH copulas using the maximum pseudo-likelihood method and corresponding AIC values. The given dataset cannot easily answer the question of tail dependence, so the upper tail-dependence coefficient, $\hat{\lambda}_U^{CFG}$, of the copulas estimated with the CFG estimator [43] supports the quantification of the tail dependence of extreme values. The $\hat{\lambda}_U^{CFG}$ value of the empirical copula is 0.582, slightly smaller than the $\hat{\lambda}_U^{CFG}$ values calculated straight from the three candidate GH copulas (Table 4). The two indicators' values illustrated the good performance of the two-parameter GH copula.

**Table 4.** Estimated parameters; two indicators: AIC and $\hat{\lambda}_U^{CFG}$ of the copulas.

| Copula Model | Parameter Name | Estimated Parameter | AIC | $\hat{\lambda}_u^{CFG}$ |
|---|---|---|---|---|
| Symmetric GH Copula | $\theta$ | 2.7576 | −519.544 | 0.714 |
| Two-parameter GH Copula | $[\beta_1, \beta_2]$ | [2.1241, 0.7176] | −520.997 | 0.614 |
| Asymmetric GH Copula | $[\theta, \pi_2, \pi_3]$ | [2.7569, 1.0000, 1.0000] | −515.480 | 0.714 |

Further, the Pickand's dependence function [44] was employed to check graphically the GOF of the candidate copulas (Figure 5). It turns out that the two-parameter GH copula fits better than the symmetric and asymmetric ones. On the basic of all the assessment indicators, the two-parameter GH (GH2) copula should be chosen to reflect the dependence structure of $W_{30d}$ at each sub-basin. Furthermore, GH2 possesses its own unique features to characterize the high relevance between extraordinary flood volumes at the two study sub-basins.

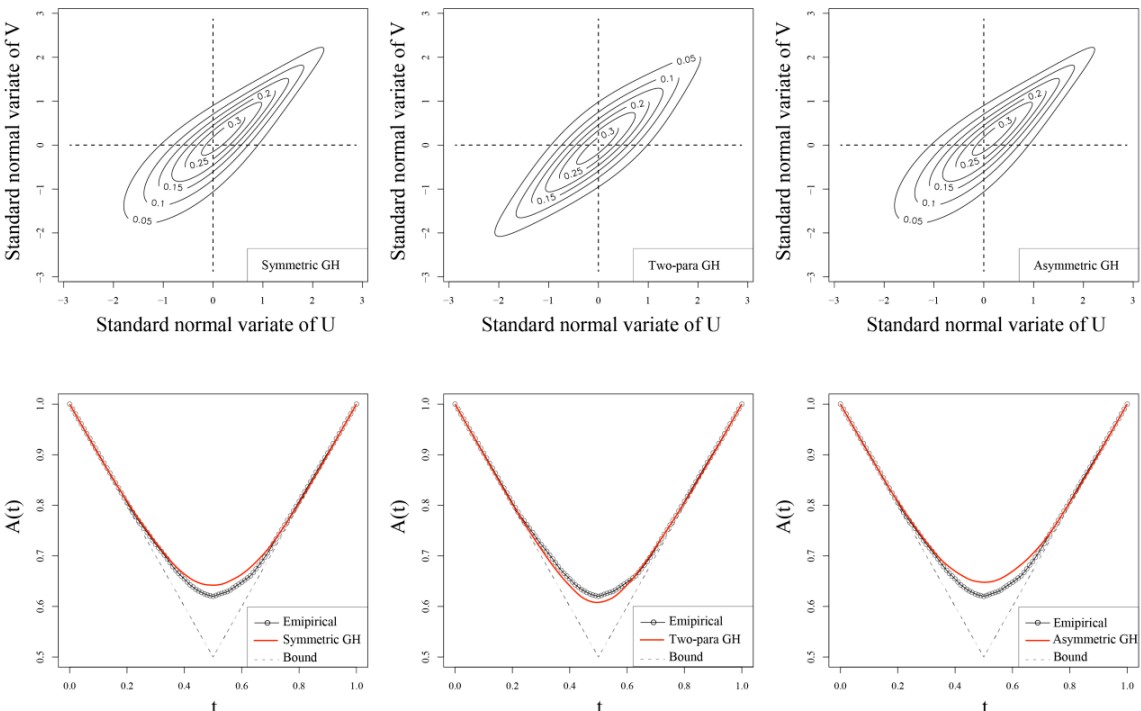

**Figure 5.** Contours of copula densities scaled to the standard normal distribution (N(0,1)) margins and plots of empirical Pickand's dependence function compared with three fitted ones.

### 4.3. CEC and CMLC Point Identification

Two copula-based regional design flood compositions (i.e., CEC and CMLC) were utilized to calculate the appropriate combinations of $W_{30d}$ that occurred at the Zhengyangguan section and Zheng-Beng interval, which have statistical basis and could satisfy the inherent disciplines of hydrologic events. For the convenience of subsequent analysis of the uncertainty of marginal distribution selection, an explored experiment combining different margins was carried out. Table 5 lists 13 analyzed combinations.

**Table 5.** Experimental design for CEC and CMLC point identification considering the uncertainty of marginal distribution selection.

| Combination | Copula | Zhengyuangguan $W_{30d}$ Distribution | Zheng-Beng Interval $W_{30d}$ Distribution |
|:---:|:---:|:---:|:---:|
| C1 | GH2 | PE3 | PE3 |
| C2 | GH2 | PE3 | LN3 |
| C3 | GH2 | PE3 | GEV |
| C4 | GH2 | PE3 | GP |
| C5 | GH2 | PE3 | GAM |
| C6 | GH2 | PE3 | GUM |
| C7 | GH2 | PE3 | GLO |
| C8 | GH2 | LN3 | GEV |
| C9 | GH2 | GEV | GEV |
| C10 | GH2 | GP | GEV |
| C11 | GH2 | GAM | GEV |
| C12 | GH2 | GUM | GEV |
| C13 | GH2 | GLO | GEV |

When the design value occurred at the Zhengyuanguan section, and the corresponding conditional expected value and conditional most likely value occurred at the Zheng-Beng interval, PE3 was applied for Zhengyuanguan $W_{30d}$, while PE3, LN3 GEV, GP, GAM, GUM, and GLO were applied for the Zheng-Beng interval $W_{30d}$ for the uncertainty analysis, which corresponds to the combinations,

C1–C7. Analogously, when the design value occurred at the Zheng-Beng interval, the corresponding conditional expected value and conditional most likely value occurred at the Zhengyuangguan section, MEV was selected as a distribution for the Zheng-Beng interval $W_{30d}$, PE3, LN3, GEV, GP, GAM, GUM, and GLO were selected as distributions for Zhengyuangguan $W_{30d}$, which corresponded to the combinations, C3, C8–C13. Three candidate marginal distributions all passed the Cramér–von Mises test given $\alpha$ = 0.05. Then, GH2 was applied to estimate the CEC and CMLC points. Table 6 lists the CEC and CMLC points determined under different combinations given T = 20, 50, 100 years using Equations (2) and (4) in the study sub-basins. For example, when a 20-year design flood occurs at the Zhengyangguan section, the CEC and CMLC points under C3 are (199.22 $\times 10^8$ m$^3$, 47.10 $\times 10^8$ m$^3$) and (199.22 $\times 10^8$ m$^3$, 48.26 $\times 10^8$ m$^3$), respectively.

*4.4. Uncertainty Analysis*

4.4.1. Uncertainty Due to the Selection of Margins

To figure out the sensibility of univariate distribution selection for CEC and CMLC point estimation, the explored experiment mentioned above was carried out: Three values of the univariate return period (T) for Zhengyuanguan $W_{30d}$ and Zheng-Beng $W_{30d}$ were selected, respectively, ranging from the moderate flood volume standard to the extreme one. The estimated CEC and CMLC points and their corresponding 90% CIs under different combinations are exhibited along with the observed data in Figure 6. By comparing these bivariate design realizations, it is expected that the impact of marginal distribution uncertainty will be discovered.

In each case shown in Figure 6, the uncertainty caused by the selection of marginal distributions increases with T. For instance, it can be shown from Figure 6 and Table 6 that when a 20-year flood occurs at the Zhengyuanguan section, the corresponding conditional expected value occurs at the Zheng-Beng interval (Case 1) ranges from 45.56 $\times 10^8$ m$^3$ to 47.55 $\times 10^8$ m$^3$, with the variation ratio of 3.48% compared with C3, while given T = 100 years, the corresponding value ranges from 63.05 $\times 10^8$ m$^3$ to 75.46 $\times 10^8$ m$^3$, with a reduction ratio of 13.54%. This phenomenon results from the CEC algorithm and the different fitting performance of selected marginal distributions. Equation (2) indicates that large conditional expected values correspond to large values of cumulative probability. When the cumulative probability of the Zhengyuangguan $W_{30d}$ event ranges from 0.95 to 0.99, the differences between the fitting performances of three margins increase with the increasing values of the cumulative probability. In Case 1, the corresponding cumulative probability of the conditional expected value of $W_{30d}$ at the Zheng-Beng interval given T = 100 years (0.9819–0.9854) is larger than that given T = 20 years (0.9235–0.9333), so the amplitude of variation of the former corresponding conditional expected value is larger than the latter. Similar analysis can be conducted for the CMLC scheme. In Case 2, the corresponding cumulative probability of the conditional most likely value of $W_{30d}$ at the Zheng-Beng interval given T = 100 years ranges from 0.9851 to 0.9876 compared with the ranges from 0.9332 to 0.9370 given T = 20 years. The finding is in line with that of Guo et al. [45] and Zhao et al. [15].

In addition, the finding implies that the marginal distributions' selection uncertainty has a slightly larger impact on the CEC scheme than the CMLC scheme. Actually, there is a comparatively small difference in all cases if any candidate distribution is used as a marginal distribution for $W_{30d}$ at the two sub-basins. Similar results can be found in the rainfall frequency analysis, which implies the negligible impact of different marginal distributions. This is rational owing to the relative errors of the performances among the seven candidate univariate distributions being less than 20%, at least for the univariate return periods regarded in the conducted experiment.

**Table 6.** CEC and CMLC points under different combinations of marginal distribution.

| Combination | | Conditional Design Regional Flood Composition Points | | | | | |
| | | T = 20 | | T = 50 | | T = 100 | |
| | | CEC | CMLC | CEC | CMLC | CEC | CMLC |
|---|---|---|---|---|---|---|---|
| Given flood occurs at the Zhengyuanguan section | C1 | (199.22, 47.55) | (199.22, 48.98) | (244.62, 59.23) | (244.62, 62.01) | (278.18, 68.39) | (278.18, 71.76) |
| | C2 | (199.22, 47.34) | (199.22, 48.37) | (244.62, 60.34) | (244.62, 61.66) | (278.18, 71.27) | (278.18, 73.34) |
| | C3 | (199.22, 47.10) | (199.22, 48.26) | (244.62, 60.83) | (244.62, 61.00) | (278.18, 72.92) | (278.18, 73.72) |
| | C4 | (199.22, 47.70) | (199.22, 50.37) | (244.62, 58.26) | (244.62, 62.14) | (278.18, 65.94) | (278.18, 70.15) |
| | C5 | (199.22, 46.50) | (199.22, 48.18) | (244.62, 56.85) | (244.62, 59.57) | (278.18, 64.83) | (278.18, 67.97) |
| | C6 | (199.22, 45.56) | (199.22, 46.91) | (244.62, 55.38) | (244.62, 57.68) | (278.18, 63.05) | (278.18, 65.81) |
| | C7 | (199.22, 46.47) | (199.22, 46.67) | (244.62, 61.34) | (244.62, 59.15) | (278.18, 75.46) | (278.18, 73.28) |
| Given flood occurs at the Zheng-Beng interval section | C3 | (178.98, 52.27) | (181.18, 52.27) | (220.24, 68.13) | (230.76, 68.13) | (252.20, 81.69) | (264.54, 81.69) |
| | C8 | (178.56, 52.27) | (180.98, 52.27) | (223.53, 68.13) | (229.34, 68.13) | (260.37 81.69) | (269.52, 81.69) |
| | C9 | (178.19, 52.27) | (179.39, 52.27) | (225.20, 68.13) | (228.18, 68.13) | (264.89, 81.69) | (271.74, 81.69) |
| | C10 | (179.48, 52.27) | (186.67, 52.27) | (213.83, 68.13) | (229.82, 68.13) | (237.23 81.69) | (253.09, 81.69) |
| | C11 | (177.86, 52.27) | (184.30, 52.27) | (217.72, 68.13) | (228.17, 68.13) | (248.45, 81.69) | (260.56, 81.69) |
| | C12 | (174.16, 52.27) | (179.31, 52.27) | (211.80, 68.13) | (220.63, 68.13) | (241.24, 81.69) | (251.81, 81.69) |
| | C13 | (176.10, 52.27) | (175.22, 52.27) | (228.12, 68.13) | (223.65, 68.13) | (276.11, 81.69) | (272.51, 81.69) |

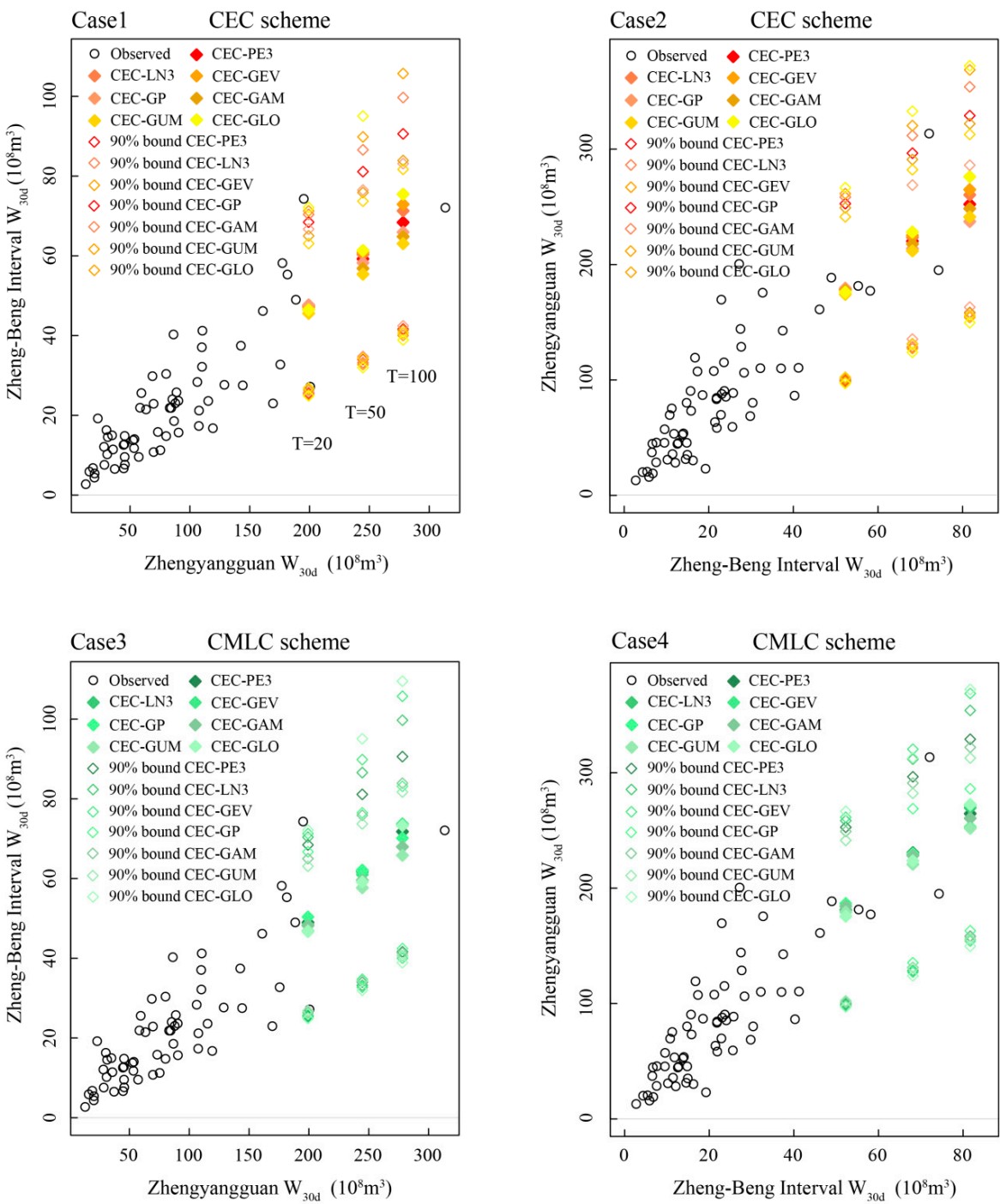

**Figure 6.** CEC and CMLC points under combinations of specified marginal distributions. Case 1 and Case 3 denote the CEC and CMLC points in the condition that the design flood (T = 20, 50, 100) occurs at the Zhengyuanguan section, respectively, while Case 2 and Case 4 denote the CEC and CMLC points in the condition that the design flood occurs at the Zheng-Beng interval, respectively. For better visualization, the horizontal and vertical coordinates of Case 2 and Case 4 have been exchanged respectively.

### 4.4.2. Sampling Uncertainty Caused by the Limited Records

For the purpose of illustrating the effect of sample sizes on CEC and CMLC point identification uncertainty, the following bootstrapping experiment was carried out under different sample sizes for 10,000 times.

1.  Three values of T for the $W_{30d}$ at two sub-basins, respectively, are selected (viz., T = 20, 50, 100 years), ranging from the moderate flood volume standard to the extreme one. Similarly, three values of sample size (sz) are selected (viz., sz = 63, 200, 500).

2.  Here, the selected model combination is C3, listed in Table 5. The CEC and CMLC events for C3 are estimated by fixing the value of sz (or T), and leaving the other variable vary in the corresponding subset. A triple of BCIs (viz., 25%, 50%, 75%) is exhibited under different schemes (sz, T). The larger the BCIs, the greater the uncertainty, and vice versa.

3.  To judge the plausibility and compare the performance of the two proposed composition methods, the contours of several selected joint probability levels [11] (viz., p-level = 0.99, 0.98, 0.95, 0.90, 0.80) together with the observed data are plotted on the same graphs as a reference.

4.  Five indexes mentioned above are also calculated (Table 7) to evaluate the sampling uncertainty of the 36 schemes.

The visual comparisons of the BCIs for different schemes are shown in Figures 7–10, for the fixed sz and T, respectively, and the three corresponding BCIs. Apparently, the variation rules of BCIs drawn in the four figures are quite similar. As expected, the BCIs significantly decrease with growing sz (row-wise) and increase with growing T (column-wise). Specifically, consider a 100-year flood event occurs in the Zhengyangguan section, meanwhile, the corresponding conditional expected value occurs in the Zheng-Beng interval, as shown in Figure 7 (top-right); the corresponding BCIs could cover p-level curves from 0.90 to 0.99, at least. Furthermore, T for the corresponding values of the Zheng-Beng interval $W_{30d}$ ranges from 150 to almost 1000 years. Evaluations of the sampling uncertainty also exists in previous studies, such as quantifying the uncertainty of hydrological droughts [15], estimating sampling uncertainty in bivariate flood quantiles estimation [34], and handling the overlap problem of the return periods of the bivariate design events [45]. As the findings of this study and previous research suggests, the large uncertainty was unable to be reduced with the observed dataset and poses a huge challenge for basin development, reservoir design, etc. [26], particularly for the Huai River basin, with high densities in both population and water projects.

In spite of the similarity of the four figures, it is difficult to quantify the uncertainty in different schemes by visual assessment. Therefore, five uncertainty evaluation indexes were employed (Table 7). With the sample size increasing from 63 to 500, the five indicators decreased by 64.1% to 88.3% under the given return periods, which conforms to the analyzed results above. Furthermore, it is exhibited in Table 7 that the values of the five metrics increase remarkably with the rising T, indicating that the sampling uncertainty induced by the limited records has greater effects for extreme floods than that for moderate and small floods.

By comparing Figure 7 with Figure 9 (or Figure 8 with Figure 10), we can find that the BCIs are a combination of two uncertainty sources: Regional design flood composition method selection and sampling uncertainty. The former ($Un_1$) quantifies the different performance of the methods proposed, while the latter ($Un_2$) describes the uncertainty related to the limited size of data, including the univariate estimation uncertainty. $\sigma_x$ and $\sigma_y$ are the uncertainty estimation indexes for one-dimensional space. When a given flood occurs at the Zhengyangguan section, the role played by $\sigma_x$ and $\sigma_y$ seem to measure $Un_2$ and $Un_1$, respectively, and the contrary is the case that a given flood occurs at the Zheng-Beng interval. However, the CEC and CMLC realizations are exhibited in a two-dimensional plane, $Un_1$ and $Un_2$ together make up the overall sampling uncertainty, and the two-dimensional indexes (i.e., $S_{25\%}$, $S_{50\%}$, $S_{75\%}$) should be better utilized. The five uncertainty metrics for the CMLC method and their variation ratio contrasted with the results of the CEC method with a resampling size of 63 are also presented in Table 7. No matter which sub-basin a given flood occurs in, the CMLC realizations have a smaller uncertainty (the corresponding variation ratios are negative) than CEC realizations when the return period of the given flood is 50 or 100, but in the case that a 20-year flood occurs, the CEC realizations seem to perform better (the corresponding variation ratios are positive). The slight difference between the listed metrics indicates that the CMLC method performs more stably

and satisfactorily for large floods, while in considered moderate and small floods, the CEC method is a better choice for uncertainty reduction.

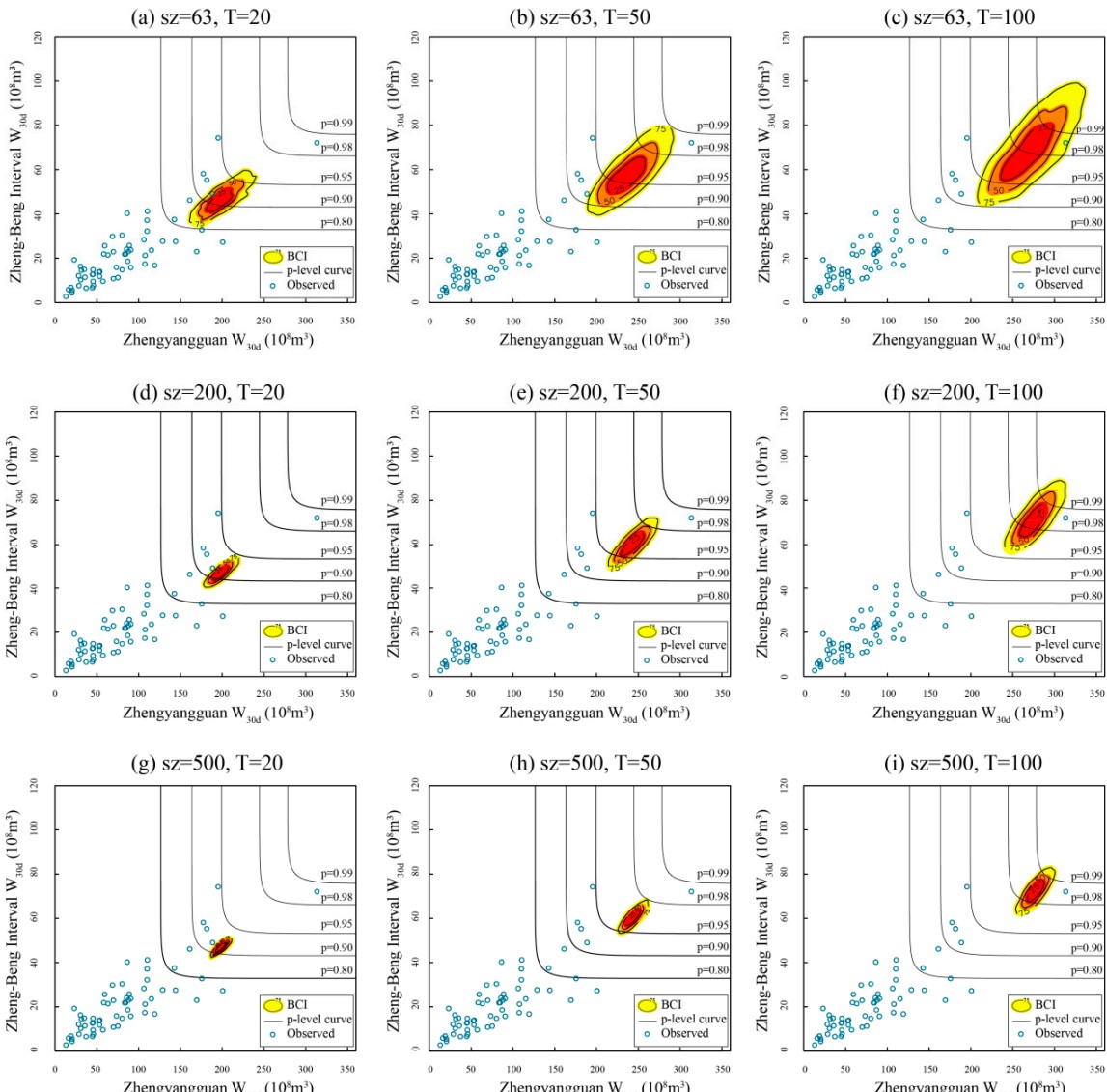

**Figure 7.** Comparison of CEC realizations under different schemes when the designed values occur in the Zhengyangguan section.

The uncertainty evaluation proposed above highlights the uncertainty caused by the limited records, which is recognized as the major uncertainty source compared with marginal selection. Indeed, there are many other possible uncertainty sources, some of which have been regarded as being insignificant, such as the copula function selection and parameter estimation method selection [25]. Except for these uncertainty sources discussed above, efforts are supposed to be made to consider additional uncertainty sources in further research. For example, the uncertainty due to different bootstrapping methods or different composition schemes other than CEC and CMLC. Additionally, more notably, with climate change, water conservancy construction, and urbanization, although the temporal distribution of flood variables and the spatial correlation between flood events that have occurred at different sub-basins may change to some degree, it is certainly worthwhile to integrate the uncertainty analysis of the regional design flood composition estimation with nonstationary flood frequency analysis [7,8] and the utilization of copula functions with time-varying parameters.

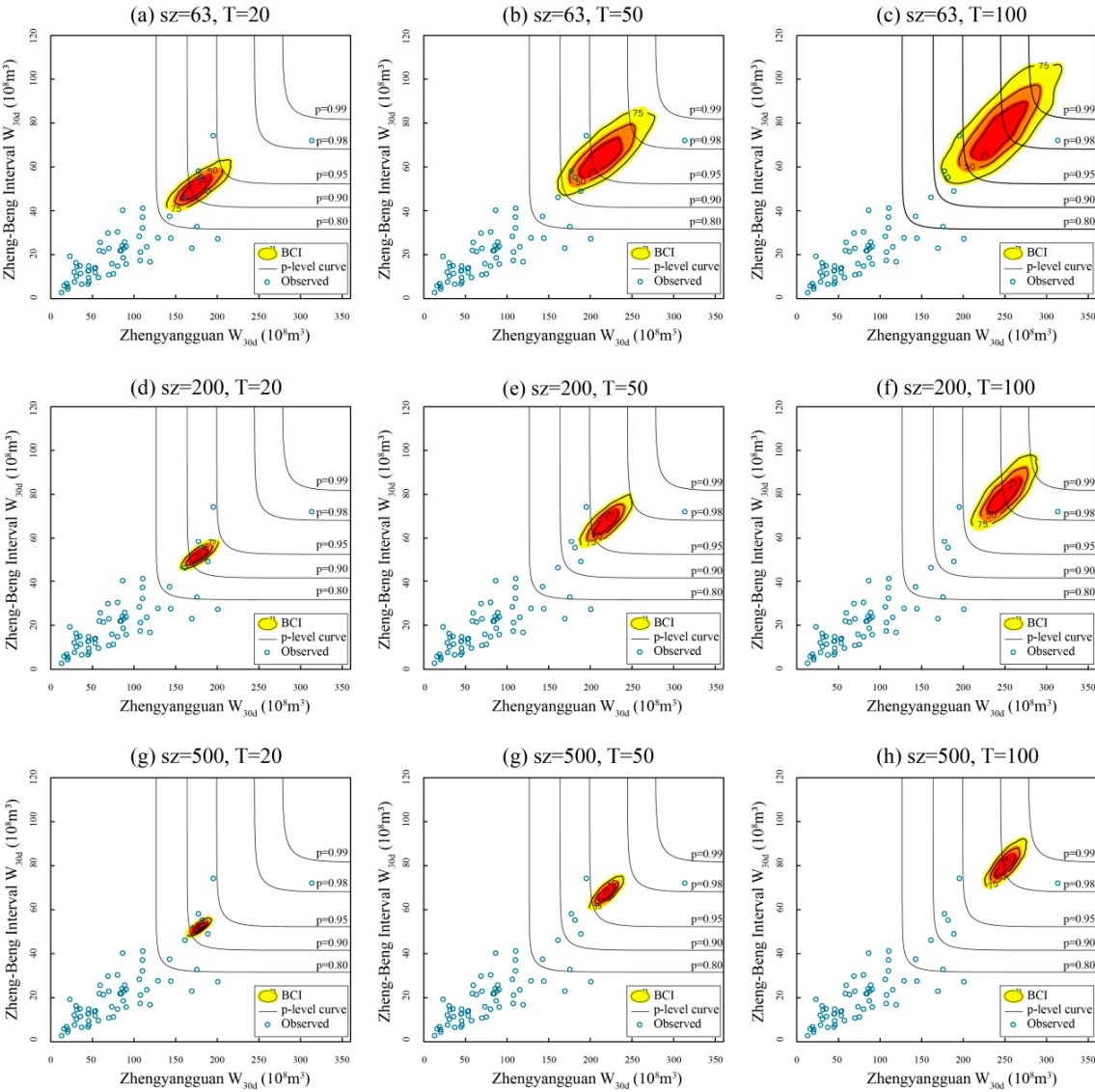

**Figure 8.** Comparison of CEC realizations under different schemes when the designed values occur in the Zheng-Beng interval.

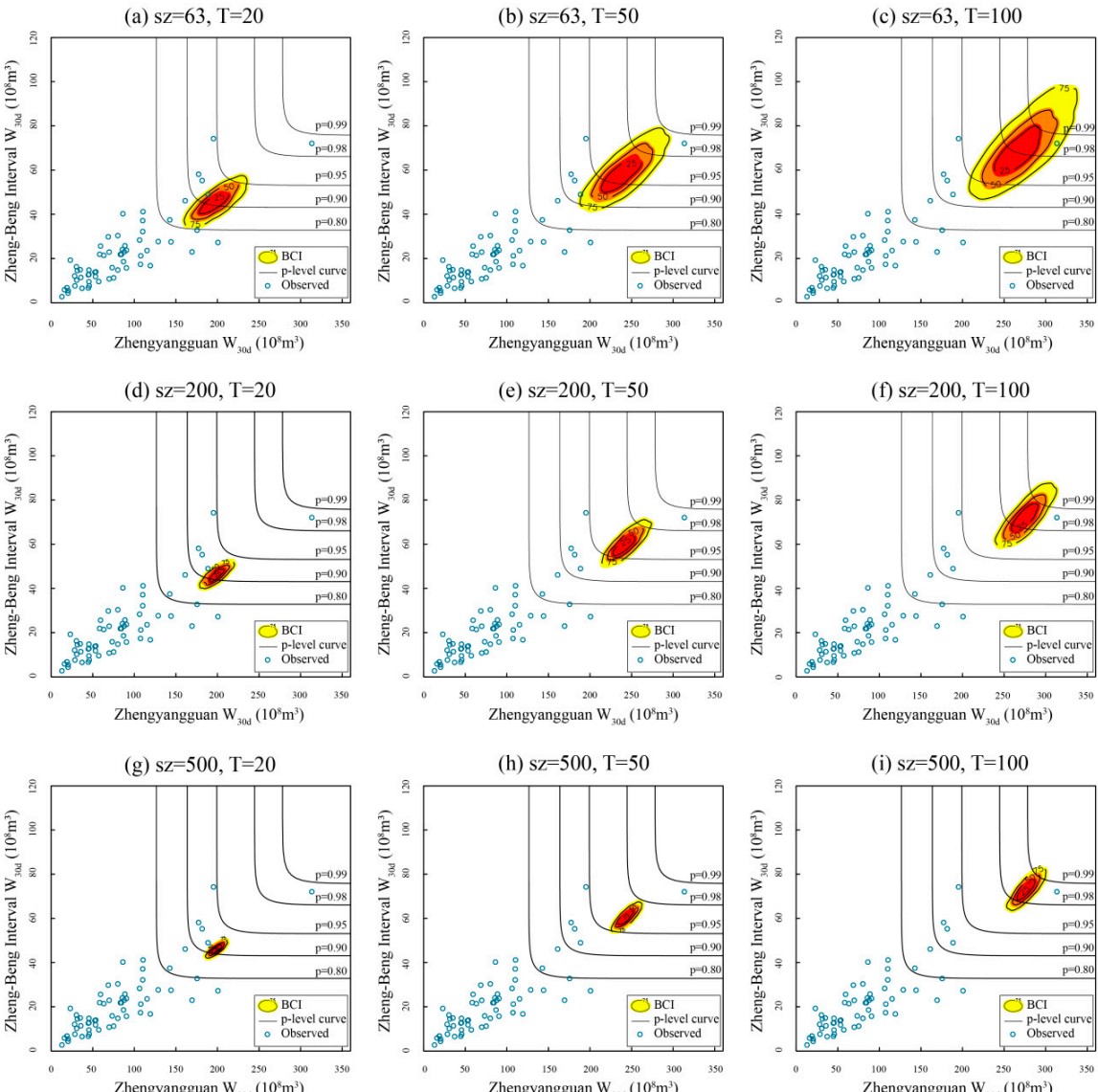

**Figure 9.** Comparison of CMLC realizations under different schemes when the designed values occur in the Zhengyangguan section.

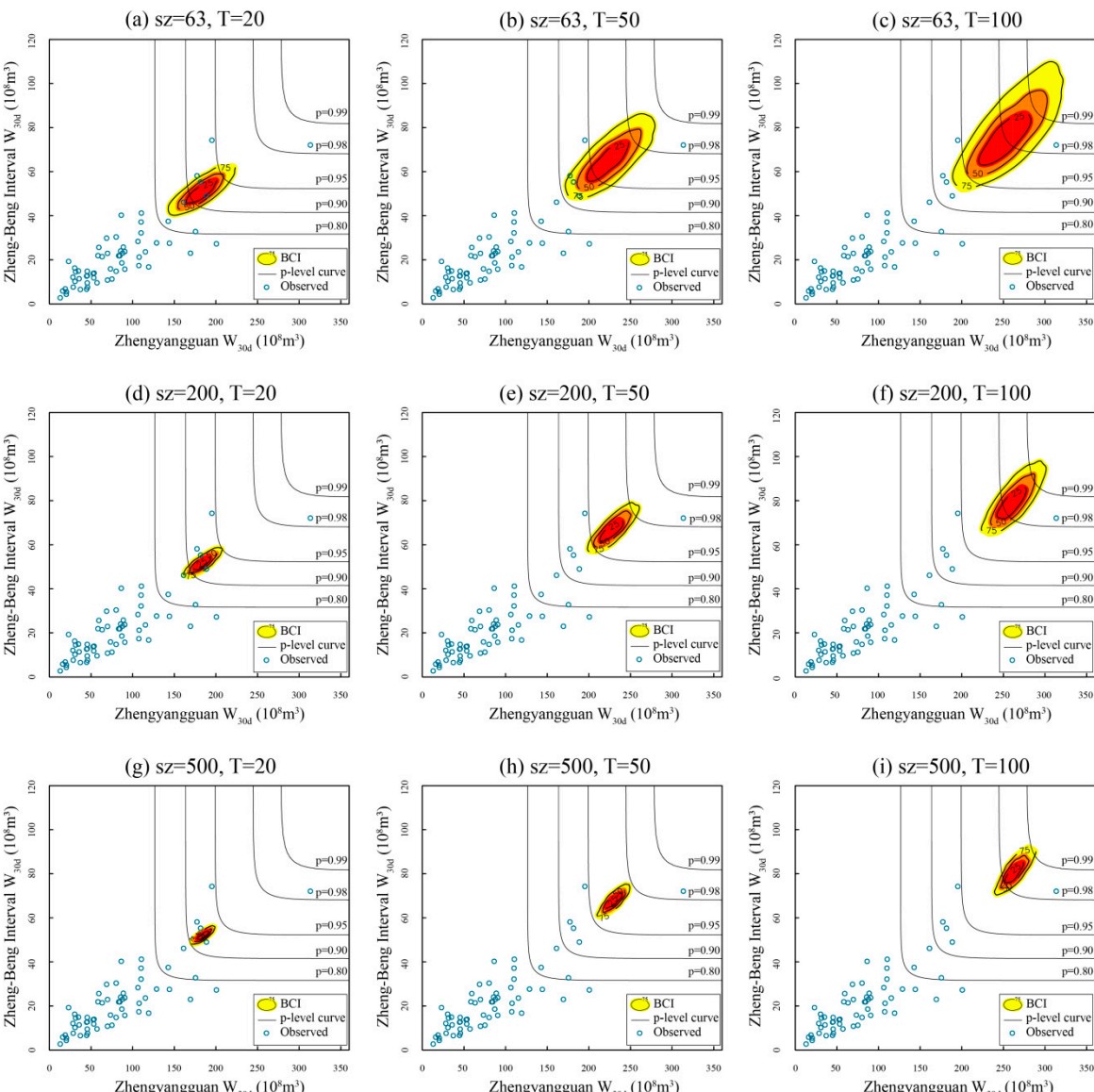

**Figure 10.** Comparison of CMLC realizations under different schemes when the designed values occur in the Zheng-Beng interval.

**Table 7.** Values of evaluation indexes for sampling uncertainty under different schemes.

| | T | Method | sz | $\sigma_x(10^8\ m^3)$ | $\sigma_y(10^8\ m^3)$ | $S_{25\%}\ (10^{16}\ m^3{\cdot}m^3)$ | $S_{50\%}\ (10^{16}\ m^3{\cdot}m^3)$ | $S_{75\%}\ (10^{16}\ m^3{\cdot}m^3)$ |
|---|---|---|---|---|---|---|---|---|
| Given flood occurs in the Zhengyangguan section | 20-year | CEC | 63 | 25.2 | 8.05 | 165 | 417 | 852 |
| | | | 200 | 12.5 | 4.06 | 50.4 | 122 | 251 |
| | | | 500 | 7.91 | 2.64 | 21.5 | 49.9 | 102 |
| | | CMLC | 63 | 25.4 (0.79%) [1] | 7.77 (−3.48%) | 162 (−1.82%) | 422 (1.20%) | 858 (0.70%) |
| | | | 200 | 12.5 | 3.85 | 49.7 | 124 | 251 |
| | | | 500 | 7.93 | 2.48 | 20.7 | 50.6 | 99.9 |
| | 50-year | CEC | 63 | 32.2 | 12.4 | 397 | 957 | 1946 |
| | | | 200 | 17.7 | 7.05 | 130 | 313 | 631 |
| | | | 500 | 11.1 | 4.45 | 51.8 | 124 | 254 |
| | | CMLC | 63 | 31.9 (−0.93%) | 11.8 (−4.83%) | 373 (−6.05%) | 891 (−6.90%) | 1866 (−4.11%) |
| | | | 200 | 17.8 | 6.37 | 116 | 281 | 564 |
| | | | 500 | 11.1 | 4.01 | 48.9 | 112 | 226 |
| | 100-year | CEC | 63 | 40.1 | 18.2 | 679 | 1656 | 3369 |
| | | | 200 | 21.9 | 10.1 | 232 | 563 | 1146 |
| | | | 500 | 13.9 | 6.41 | 94.2 | 230 | 462 |
| | | CMLC | 63 | 40 (−0.25%) | 16.7 (−8.24%) | 626 (−7.81%) | 1537 (−7.19%) | 3301 (−2.02%) |
| | | | 200 | 21.6 | 8.93 | 199 | 489 | 967 |
| | | | 500 | 13.9 | 5.74 | 85.2 | 201 | 409 |
| Given flood occurs in the Zheng-Beng interval | 20-year | CEC | 63 | 23.8 | 6.86 | 166 | 405 | 812 |
| | | | 200 | 13.4 | 3.91 | 56.1 | 133 | 262 |
| | | | 500 | 8.55 | 2.51 | 22.8 | 53.9 | 108 |
| | | CMLC | 63 | 24.6 (3.36%) | 6.95 (1.30%) | 174 (4.82%) | 410 (1.23%) | 846 (4.19%) |
| | | | 200 | 13.8 | 3.94 | 55 | 137 | 275 |
| | | | 500 | 8.6 | 2.51 | 21.7 | 53.1 | 109 |
| | 50-year | CEC | 63 | 34.1 | 12.2 | 420 | 1018 | 2106 |
| | | | 200 | 19.3 | 7.03 | 144 | 341 | 684 |
| | | | 500 | 12.1 | 4.41 | 56.9 | 137 | 280 |
| | | CMLC | 63 | 34 (−0.29%) | 12.3 (0.82%) | 405 (−3.57%) | 997 (−2.06%) | 2073 (−1.56%) |
| | | | 200 | 18.4 | 6.87 | 134 | 317 | 631 |
| | | | 500 | 11.6 | 4.43 | 56.6 | 129 | |
| | 100-year | CEC | 63 | 42.2 | 18.2 | 739 | 1793 | |
| | | | 200 | 23.9 | 10.2 | 260 | 638 | |
| | | | 500 | 14.9 | 6.41 | 102 | 253 | |
| | | CMLC | 63 | 41.9 (−0.72%) | 18.2 (0.00%) | 712 (−3.65%) | 1780 (−0.73%) | |
| | | | 200 | 22.6 | 10.3 | 250 | 599 | |
| | | | 500 | 14.1 | 6.58 | 101 | 235 | |

[1] Values shown in parenthesis are variation ratios compared to the CEC method.

## 5. Conclusions

It is argued that regional design flood composition is in general a spatially stochastic problem, and construction of the joint distribution of flood variables at each sub-basin is the most reasonable approach to solve this issue. Bengbu hydrological station was selected as a case study, and natural $W_{30d}$ series at its two sub-basins (i.e., Zhengyangguan section, Zheng-Beng interval) were obtained. The copula function was adopted to capture the dependence structure of $W_{30d}$ at each sub-basin for its flexibility to select any complex marginal distributions. For the purpose of integrated river-basin development, the CEC and CMLC methods were utilized to obtain the design flood combinations. To figure out the sensibility of marginal distribution selection and the influence of sampling uncertainty caused by the limited records on the two composition methods, a corresponding comprehensive experiment was designed. Here, to clarify the former uncertainty source, seven candidate univariate distributions were combined according to certain rules to produce 13 combinations for fitting the $W_{30d}$ series at each sub-basin. For the quantification of the latter uncertainty source, the CC-PB procedure was developed and five evaluation indexes were calculated. The following conclusions have been drawn from this study:

We recommend the PE3 distribution for modeling of the marginal of $W_{30d}$ at the Zhengyangguan section, and MEV for the $W_{30d}$ at the Zheng-Beng interval. Taking the overall dependence structure and tail dependence into account, GH2 was awarded as the most appropriate model.

The CEC and CMLC realizations were estimated using the constructed model. The experiment results indicate that the CEC and CMLC point identification has a close relationship with the fitting performances of the three selected univariate distributions. Namely, the smaller fitting performance distinction amongst distribution choices under a specific cumulative probability, the closer the location of the corresponding estimated CEC and CMLC points. Therefore, the uncertainty resulting from marginal distributions' selection needs to be properly recognized.

The sampling uncertainty caused by the limited records was considerably high, which should arouse more attention. The CEC and CMLC points under a specific univariate return period showed significant variation. The $S_{75\%}$ of two regional design flood composition events covered more than one p-level curve. Consequently, it leads to an undervaluation or overvaluation of the risk related to hydrological designs [23].

Both analyzed sources of uncertainty increased with the growing T. As for the comparison of the two proposed methods, it seemed that the uncertainty due to the marginal selection had a very slight larger impact on the CEC scheme than the CMLC scheme; but in terms of sampling uncertainty, the CMLC method performed slight stably for large floods, while in considered moderate and small floods, the CEC method performed better. The comprehensive uncertainty analysis indicates that considerable uncertainty is accompanied with the process of CEC and CMLC point identification, which is inevitable to some extent, while the findings in this research can provide directions and suggestions to flood risk reduction and basin development. On the one hand, appropriate univariate distribution is important for regional design flood composition event identification and it is essential to improve the fitting accuracy of flood variables by improving the parameter estimation performance of the candidates or constructing new distributions using information entropy [45]. On the other, it is far from sufficient to constrain the sampling uncertainty even for a sample size larger than 50 (63 years of observed data). It is necessary to further expand the information content beyond the flood records [46], such as combing historical flood data, conducting regional flood frequency analysis, or deriving a flood frequency approach. Besides, the confidence interval information contained in two copula-based compositions provides a basis to clarity the flood risk.

Overall, our research emphasizes the significance of appreciating the uncertainty gone with the process of copula-based regional design flood composition estimation. It is worthwhile to consider other possible uncertainty sources and the impact of climate change and human activities on CEC and CMLC point identification in future investigations.

**Author Contributions:** S.M., P.S. and Y.F. proposed the methods and developed the model; X.J. and L.Z. did computation and simulation work; S.M., C.C. and F.D. analyzed the results; S.M., P.S. and S.Q. wrote the paper.

**Funding:** This research is financially supported by the National Key Research and Development Plan (2017YFC0405601), the National Natural Science Foundation of China (No. 41730750, 51479062), the UK-China Critical Zone Observatory (CZO) Program (41571130071).

**Conflicts of Interest:** The authors declare no conflict of interest.

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
