# Peer review of "Uncertainty Analysis of Two Copula-Based Conditional Regional Design Flood Composition Methods: A Case Study of Huai River, China"

_water, doi:10.3390/w10121872_

Round 1
Reviewer 1 Report
COMMENT 1
Why did the authors consider only three candidata distributions PE3, LN3, GEV? The authors should use at least 5 distributions. For instance, apart from PE3, LN3, GEV, other distributions that the authors need to consider include the Generalize Pareto distribution (GPD), Gamma distribution, Gumbel distribution, Generalized Logistic distribution, etc.
COMMENT 2
A brief review on studies which used copula-based approach of estimation of regional flood
designs should be given.
COMMENT 3
The authors need to link their paper's analysis to papers that have previously appeared in
Water-MDPI. This is in line with a question of relevancy, i.e., whether Water-MDPI readers will
benefit maximally by a paper that may not strongly posses evidence of an impact and above all
relevancy to the journal's contents.
COMMENT 4
What are the limitations of this study? How can such limitations be addressed in future research?
COMMENT 5
The authors should compute some statistical "goodness-of-fits" especially the Akaike Information
Criteria to show the performance of the candidate distributions.
Author Response
We are thankful to the reviewer for the thoughtful and constructive comments that we believe has helped us to improve the manuscript. Following the reviewer's suggestion, we revised every problem proposed by the reviewer as outlined in the responses attached.(water-405094-response to reviewer 1.doc)

Reviewer 2 Report
The paper presents study on sensibility of marginal distribution selection and the impact of sampling uncertainty caused by the limited records on two copula-based conditional regional design flood composition method. Research is carried out basing on case study of Huai River in China. The paper can be publish in Water journal after minor improvements:
Improper citation style (Harvard mixed with MDPI style). Please follow instructions for Authors.
Some caption names in bibliography [pos. 2]
Tab. 1: All symbols used in functions must be described
I strongly suggest to add research/procedure workflow diagram
I really do not understand the meaning of fig.1a and the reason You included this visualisation. There is no introduction to it in the text also. If Bengbu is closing cross section why did You visualised the area which is not study site? [the right part of the map]
General remark: graphics and charts should be much larger! Please make them readable.
There is no discussion section. Please add it before conclusions. You can also provide combined section. I also suggest to update the literature review. Take some papers from yrs 2017-2018. You declared [line 53] that “In recent years, the copula function has been extensively used” please provide references, and then discuss them.
Could You expand the final thought This research provides new approaches for analysis of regional design flood composition and useful information to clarity flood risk? Please add some findings which can be applicable in practise.
Author Response
We are thankful to the reviewer for the constructive and encouraging comments that we believe has helped us to improve the manuscript. Following the reviewer's suggestion, we revised every problem proposed by the reviewer as outlined in the responses attached.

Round 2
Reviewer 1 Report
The paper has been greatly improved upon except the minor editorial changes which can be made. The paper can now be accepted for publication. The minor changes are as below:
COMMENT 1
The authors should harmonize the styles of referencing e.g. Yan et al. (2010), Guo et al. (2018), [#1], [#2],.....
COMMENT 2
Line 121:
Replace:
"dependence. Gumbel-Hougaard (GH)"
with
"dependence. Another set of extreme events which would be of interest (though not considered in this study) comprise peak volumes occurring in an independent and identically distributed way or the peak-over-threshold events the extraction of which follow the procedure like that found in Onyutha (2017). Gumbel-Hougaard (GH)"
Onyutha C (2017) On Rigorous Drought Assessment Using Daily Time Scale: Non-Stationary Frequency Analyses, Revisited Concepts, and a New Method to Yield Non-Parametric Indices, Hydrology 2017, 4(4), 48; doi:10.3390/hydrology4040048
COMMENT 3
Line 492: change "Both the two analyzed" to "Both of the analyzed"
Author Response
We are thankful to the reviewer for the constructive and encouraging comments that we believe has helped us to improve the manuscript. Following the reviewer's suggestion, we revised every problem proposed by the reviewer as outlined in the responses below.
